# *Burkholderia cepacia* Complex Infections in Urgently Referred Neonates from Syrian Border Regions to a Hospital in Turkey: A Cross-Border Cluster

**DOI:** 10.3390/children9101566

**Published:** 2022-10-16

**Authors:** Benhur Sirvan Cetin, Ayşen Orman

**Affiliations:** 1Division of Pediatric Infectious Diseases, Department of Pediatrics, Faculty of Medicine, Erciyes University, Kayseri 38039, Turkey; 2Division of Neonatology, Department of Pediatrics, Faculty of Medicine, Mersin University, Mersin 33110, Turkey

**Keywords:** *Burkholderia cepacia* complex, neonate, bacteremia, sepsis, Syria, refugees, war, outbreak

## Abstract

*Burkholderia cepacia* complex (BCC) is a rare cause of sepsis in neonates, but infections are usually severe. It can be encountered unexpectedly when adequate health care is not provided. In this study, 49 neonatal cases with blood culture-proven BCC bacteremia within the first 72 h following admission to the neonatal intensive care unit between June 2017 and December 2018 were retrospectively analyzed in detail. All but one of the cases were born in Jarabulus, Al Bab, or Aleppo in Syria and were referred to Turkey due to urgent medical treatment needs. The rate of BCC bacteremia among the neonates transferred from across the border was 16.1% (48/297). The most common coexisting problems in the cases were multiple congenital malformations (12.2%), gastrointestinal system atresia (8.2%), and congenital heart diseases (4.1%). The median age at the time of their admission in Turkey was three days, and the median length of stay in another center before the referral was 11.5 h. The case fatality rate was 14.3%. In this study, a high rate of BCC infection and associated mortality was seen in neonates referred from cross-border regions. For centers accepting cases from conflict-affected regions, it is crucial to be careful regarding early detection of bacteremia, planning appropriate treatments, and preventing cross-contamination risks within the unit.

## 1. Introduction

The *Burkholderia cepacia* complex (BCC) initially emerged in the 1980s as opportunistic human pathogens causing severe and life-threatening infections [1]. Based on molecular analysis, *Pseudomonas cepacia* has been separated from *Pseudomonas* and renamed as *B. cepaci*. It was initially thought to be one specific bacterial species. However, BCC now constitutes 24 closely related *Burkholderia* species, including *B. cepacia*, *B. multivorans*, *B. cenocepacia*, and others that can be differentiated by molecular and biochemical methods [2]. Members of BCC are aerobic, catalase-producing, Gram-negative bacteria not considered part of normal human flora [3]. They can survive for long periods in water and are intrinsically resistant to several antimicrobial agents [4,5]. Strains of the BCC are opportunistic nosocomial pathogens that can cause severe infections in children, especially with immunodeficiency, malignancy, congenital heart disease, or chronic respiratory diseases. These infections most commonly present with respiratory tract, urinary tract, and bloodstream infections and can lead to outbreaks through different sources [6]. In neonates, BCC infections are usually reported as outbreaks in the neonatal intensive care unit (NICU). In addition to nosocomial infections, very few studies in the literature show that BCC is an important infectious agent group that can also cause community-acquired infections in areas with limited access to healthcare [7,8]. In this study, neonates referred to a NICU in Turkey from different centers, including the cross-border regions of Syria, shortly after birth and who were diagnosed with BCC bacteremia at admission were examined.

## 2. Materials and Methods

This retrospective study was conducted in the NICU of the Cengiz Gökçek Maternity and Children Hospital in Gaziantep, Turkey, which is defined as a referral hospital for both neighboring provinces in Turkey and the Syrian regions. Annually, approximately 1500 neonates are admitted to the NICU with 83 incubators (56 level III, 17 level II, and 10 level I). Per the hospital policy, blood cultures are routinely taken at admission from cases referred from other centers. If necessary, samples are taken from other body fluids (such as urine, tracheal aspirate, and cerebrospinal fluid [CSF]), and blood culture is repeated during hospitalization.

This study analyzed blood culture-proven BCC infections within the first 72 h after admission between June 2017 and December 2018. The data included birthplace, gestational age, birth weight, gender, nationality, coexisting diseases, microbiological and routine laboratory (complete blood count and acute phase reactants) values, clinical profile, and outcomes and were retrieved from digital patient records. Neonates whose first positive BCC culture was obtained 72 h after hospitalization were excluded from the study.

BACTEC^TM^ Pediatric bottles (BD Diagnostics, Sparks, MD, USA) were used for blood culture. Blood samples for routine investigations and culture were obtained concurrently at admission. Blood culture samples were taken to a fully automated blood culture system BD BACTEC™ (Becton Dickinson, Sparks, MD, USA). The identification and susceptibility tests were performed by conventional methods and BD Phoenix^TM^ (BD Phoenix System, Becton Dickinson, Sparks, MD, USA) panels. Antibiotic susceptibility testing was conducted using selected antibiotics recommended for testing against BCC by the Clinical Laboratory and Standards Institute (CLSI): ceftazidime, co-trimoxazole, meropenem, and levofloxacin. Antibiotic susceptibility for BCC was interpreted as per recommendations of the CLSI [9]. Cerebrospinal fluid (CSF) analysis and culture were performed on all neonates if there was no contraindication.

Environmental swab cultures were taken from ambulances and transport incubators to control infection and find the source. The hospital in the Jarabulus region of Syria, one of the centers from which the cases were referred to us, was visited. Swab samples were obtained from potential sources in the neonatal intensive care unit and delivery room (including the water tank of incubator humidifiers, heated humidification water from respirators, tap water, incubator surfaces, intravenous solutions, and antiseptics). The samples were inoculated into BHI broth, incubated at 37 °C for 5 d, and then subcultured onto chocolate agar and MacConkey agar. Isolates, if present, were identified by standard microbiological techniques as used in the other cultures.

### Statistical Analyses

Statistical analyses were made after the data were anonymized so as not to contain personal information. The IBM SPSS Statistics for Windows version 25.0 (IBM Corporation, Armonk, New York, United States) was used for statistical analysis. The variables were investigated using visual (histogram, probability plots) and analytic methods (Kolmogorov–Smirnov, Shapiro–Wilk tests) to determine whether they were normally distributed. Descriptive analysis was presented using median (minimum [min.]–maximum [max.] values) for continuous variables and frequency (percentage) for categorical variables. Categorical data were statistically analyzed by Fisher’s exact or chi-square tests, as appropriate. Continuous data were analyzed by Student’s *t*-test or Mann–Whitney U test, whichever was appropriate. *p* values < 0.05 were considered statistically significant.

## 3. Results

### 3.1. Distribution of Patients in the Study Period

During the 19-month study period (between June 2017 and December 2018), a total of 2562 patients were hospitalized in the NICU. Of these cases, 45.5% (1165/2562) were Syrians. When all hospitalized patients are grouped according to their place of residence and nationality, Turks were 54.5% (1397/2562), Syrians residing in Turkey were 33.9% (868/2562), and those referred to our hospital from Syria were 11.6% (297/2562). Among all these patients, BCC infection with positive blood culture was detected within the first 72 h after hospitalization in 49 neonates. All of these infected neonates were Syrian. All but one of them were born in Syria, in Al Bab, Jarabulus, or Aleppo, and had to be urgently referred to our center in Turkey by ambulance, as their general condition was not good. One case was born in Kilis, a city on Turkey’s border with Syria, and was referred to our hospital for further examination and treatment. The cases’ birthplaces and the province of Gaziantep, where our hospital is located by emergency referral, are shown on the map in Figure 1. Of all BCC infected cases, 65% (32/49) were hospitalized between August 2017 and February 2018.

The number of cases decreased significantly after June 2018; the last case was seen in December 2018. The distribution of infected cases with BCC during the 19 months is shown in Figure 2 by grouping them according to their place of birth. A BCC bacteremia was observed within the first 72 h of hospitalization in 16.1% (48/297) of newborns who were urgently referred to our center in Turkey from Syria. BCC bacteremia was observed in none of the Turkish cases and Syrian neonates born in Turkey, except one admitted to our center in the same period. Among the cases referred from Syria, we saw *Acinetobacter baumannii* bacteremia in 4 cases, *Serratia marcescens* in 2 cases, *Klebsiella pneumoniae* in 1 case, and *Sphingobacterium spp.* bacteremia in 1 case. These cases were not included in further analysis because they were outside the scope of this study.

### 3.2. Characteristics of the Neonates at the Admission

The data of 49 newborns with blood culture-proven BCC infection within the first 72 h of hospitalization were analyzed in detail. A total of 67.3% (33/49) of the cases were male, and 100% (49/49) were Syrian. Places of birth were Jarabulus (49%), Al Bab (40.8%), Aleppo (8.2%), and Kilis (2%), in order of frequency. The median age at their hospitalization was three days, and the median length of stay in another center before the referral was 11.5 h. The rate of vaginal delivery was 59.2% (29/49), prematurity was 49% (24/49), and the rate of those born under normal birth weight was 57.1% (28/49). The main referral reasons of the cases were prematurity, respiratory distress, and sepsis. A total of 66.7% (33/49) of the cases had no underlying comorbidity conditions. The most common underlying comorbidities in the cases were multiple congenital malformations (12.2%), gastrointestinal system atresia (8.2%), and congenital heart diseases (4.1%). Detailed demographic and clinical features at admission are shown in Table 1. Since detailed data about the prenatal follow-ups of the cases, maternal risk factors, and the treatments they received from birth to referral to us could not be reached, an evaluation could not be made on this subject.

### 3.3. Microbiological Characteristics, Laboratory Values, Clinical Profile during the Infection, and Outcomes of the Neonates

BCC was isolated in CSF and blood in one of the 49 patients. A total of 89.8% (44/49) of positive cultures were obtained within the first day of hospitalization. All agents were sensitive to ceftazidime, levofloxacin, and meropenem, while the sensitivity of co-trimoxazole was 57.1% (28/49). The co-trimoxazole resistance rate in the causative agents was 50%, 35%, 50%, and 0% in the cases from Jarablus, Al Bab, Aleppo, and Kilis, respectively. No significant difference in co-trimoxazole resistance was found between the regions (*p* = 0.6). Microbiological characteristics, laboratory values, clinical profile during the infection, and outcomes of the neonates are shown in Table 2. A central catheter was inserted in 46.9% of the cases at the time of culture, and total parenteral nutrition was initiated in 32.7% of the cases. It was observed that the general conditions of the referred cases were generally poor at the time of admission to the center. After hospitalization, mechanical ventilation was required in 49% (24/49) of the cases, noninvasive ventilation (CPAP) in 28.6%, and inotropic infusion support in 34.7% of the cases. At the time of detection of bacteremia, CRP was higher in 85.7% of the cases, and procalcitonin value was higher than normal in 73.5% of the cases.

In our center, we were starting ampicillin and gentamicin empirically because of the high risk of infection in neonates coming from across the border. After the BCC case cluster started, we started to use carbapenem in the empirical antibiotic treatment for neonates who came from across the border and whose general conditions were not good. Empirical antibiotic treatment was discontinued in infants whose clinical and laboratory findings did not correlate with an infection. The treatment of the cases with BCC infection was continued with meropenem. In neonates with BCC infections who responded quickly to the initial treatment and did not develop additional complications, meropenem treatments were deescalated to ceftazidime, and the antibiotic treatment was completed for a total of 14 days. The median hospital stay of the cases was 25 days, and 7 cases (14.3%) died within two weeks following bacteremia. In all neonates who died, the general condition was not good, or an underlying disease was present at the admission. Their empirical antibiotic treatments were started and continued with meropenem. The characteristics of the non-survivor neonates with BCC infections are shown in Table 3. One of the non-survivors had a diaphragmatic hernia, two had congenital heart disease, one had anal atresia, and the other had multiple congenital malformations. In one of the cases, *Klebsiella spp.* co-infection was detected before she died.

### 3.4. Comparison of the Survivor and Non-Survivor Neonates with Burkholderia cepacia Complex Infections

The characteristics of survivor and non-survivor neonates were compared for all variables in Table 4. Mortality was higher in patients with an underlying comorbid disease and those who needed respiratory and inotropic support at the time of culture (*p* = 0.02, *p* = 0.01, and *p* = 0.002, respectively). There was no significant difference in mortality rates according to the region of origin of the cases. It was observed that birth weight, gestational age at birth, age at presentation, and other demographic and clinical features did not affect mortality. There was no relationship between the primary laboratory data (complete blood count and acute phase reactants) and co-trimoxazole resistance of the agent with mortality at the time the infection was first detected.

### 3.5. Assessments and Investigations for the Infection Prevention and Control

Following the initial detection of the case cluster, several studies were carried out both in and out of our center for infection prevention and control. First, cross-contamination was prevented by cohorting the cases admitted from across the border until their first cultures resulted in our unit. The culture collection and laboratory procedures at our center were reviewed. No growth was observed in the environmental swab cultures taken in the unit. Except for the cases referred from Syria and a neonate from Kilis, BCC was not detected in any patients hospitalized either in the NICU or other wards of the children’s hospital. Swab cultures were taken from the transport incubators and their surroundings in the ambulances carrying the cases. No growth was detected in the swab cultures. The hospital in Jarabulus was visited, and the employees held an information meeting on infection control measures. Swab cultures were taken from various parts of the neonatal department and delivery room in Jarabulus. *Acinetobacter baumannii* and *Klebsiella pneumoniae* were detected only in cultures taken from incubator humidifier chambers. However, BCC was not detected in any samples from Jarabulus. Hospital disinfection and sterilization procedures were reviewed, and practices were closely controlled at the Jarabulus center. The Al Bab and Aleppo centers could not be visited due to unsuitable conditions.

## 4. Discussion

BCC is an emerging pathogen, especially in children and immunocompromised patients, which can cause high morbidity and mortality [3,5,10]. The involvement of BCC in neonatal sepsis is a major concern; such infections can be acquired either from maternal flora during the intrapartum period or from elsewhere during the immediate postnatal period.

In a multicentric cross-sectional study in Yemen, *B. cepacia* (37%) was the most common microorganism causing neonatal sepsis [7]. In a neonatal sepsis surveillance study covering the years 2017–2020 in India, Gram-negative non-fermentative bacilli accounted for 29% of all causative agents, among which the most common were *A. baumannii* (37.5%) and *B. cepacia* (33.3%) [11]. It is seen that advanced molecular analyses were not performed on the strains defined as *B. cepacia* in these two studies, so it would be more appropriate to accept them as BCC. The risk of hospital-acquired infection is exceptionally high for neonates admitted to hospitals in low-resource settings and is associated with overcrowding, understaffing, and weak infection control protocols [3]. Cross-transmission, frequent pulmonary procedures, surgeries, and instrumentation, including central venous access, facilitate this organism’s nosocomial spread [5,12]. BCC can lead to outbreaks through different sources that include contaminated intravenous medications/fluids, medical devices, ultrasound probe gels, lipid emulsion stoppers, mouthwash, nebulizer solutions, or skin disinfectants [4,6,10,12,13,14,15]. In a systematic review of healthcare-associated BCC outbreaks, medical products are found to be the most frequent source of the outbreaks, representing over half of the identified sources, with 12% of the outbreaks caused by disinfectant products [16]. Maternal risk factors such as poor intrapartum and postnatal infection control practices also play a role in developing neonatal infections [3,17].

The cases referred to our study center were born in different centers and were referred due to the need for advanced medical support. When we look at the distribution of the cases, all but one came from across the border. Since blood cultures were taken during the first hospitalization, the infections detected in these cases were not originating from our center. No similar BCC infection was detected in other babies born in Turkey and admitted to the center in the same period. This indicates that the case cluster is of cross-border origin. The ambulances and teams carrying the cases were different, no growth was detected in cultures taken from ambulances, transport incubators, and other environments, and BCC was not detected in cultures taken from other neonates hospitalized in our center. All those findings excluded the possibility of pseudobacteremia and contamination. A commonly used medicinal product in all centers may be the source of the epidemic. Since both investigations for the infection source and molecular clonal typing analyses could not be performed in the centers, we could not determine whether the infections were community- or hospital-originated in Syria.

In our study, approximately half of the infected neonates were preterm (%49), and the median birth weight was 2300 g. The rate of the cases who did not have a normal birth weight was 57.1%. In the case series reported from India, Malaysia, and Paris, the cases are mostly premature and have low birth weight. [12,13,18]. In the study of Chandrasekaran et al. in India, one of the largest case series in the literature, maternal risk factors were detected in 29% of 59 infants infected with BCC, and 59% of infections developed before the third postnatal day [19]. In our study, 33.3% of our cases had many underlying diseases, such as congenital malformations (syndromic infants), congenital heart diseases, and intestinal atresia. The median age of hospitalization was also three days. Our data, combined with previous studies, suggest that infants with an underlying risk factor are at greater risk for BCC infections, even if geography and possible sources of infection differ. The Yemen study also identified vaginal delivery as an independent risk factor for neonatal sepsis [7]. When evaluated in terms of nosocomial infections, the prolonged use of central lines in neonates was suggested as a significant risk factor in an outbreak of BCC septicemia in a NICU [5]. When the cases were admitted to the center in our study, their general condition was poor, and they were already infected. Before the referral, there was a history of hospitalization for different durations in another center. Under the current conditions, we could not determine how many of the cases were community-acquired and how many were hospital-acquired, since it was impossible to fully access the health data related to the mothers and cases.

The findings are nonspecific in infections due to BCC, and neonates may present with any manifestation of bacterial sepsis. In a study examining neonatal sepsis due to non-fermentative Gram-negative bacteria, 168 neonates with sepsis commonly exhibited feeding intolerance (64.5%), a need for respiratory support (56.2%), circulatory insufficiency that necessitated vasopressor treatment (54.1%), disseminated intravascular coagulation (35.4%), and seizures (33.3%) [11]. In another study reported from India, the clinical presentation of 59 neonates infected with BCC had respiratory distress (97%), hemodynamic instability (83%), vomiting (53%), and abdominal distention (49%) [19]. In our study, 77.6% of infected neonates needed respiratory support, and 34.7% needed inotropic infusion support. TPN support was given to 32.7% of the cases due to nutritional problems and sepsis clinics.

Changes in the leukocyte and platelet counts in neonatal sepsis can be considered as a response to inflammation. However, in studies, a significant relationship is not always seen. The Yemen study found no association between the culture-positive and culture-negative groups regarding leukocyte or platelet counts [7]. In the study of Chandrasekaran et al. including 59 neonates, CRP elevation was found in 71% of the cases during bacteremia, and the median leukocyte count of the cases was 13,500 cells/mm^3^, and the platelet count was 20,000/mm^3^ [19]. While the leukocyte count was similar in our study, the platelet count was generally low in the cases. CRP and procalcitonin elevation were elevated in 85.7% and 73.5% of our cases, respectively. In the study of Patra et al., in which they examined 12 neonates who developed BCC bacteremia, growth in CSF was detected in 3 neonates [6]. CSF growth was detected in one of our 49 cases. When we evaluate all these results together, it should not be forgotten that basic haematological and inflammatory parameters may be within normal limits when diagnosing BCC infections in neonates. CSF analysis should be performed in infants unless there is a contraindication.

Appropriate antimicrobial therapy is challenging in BCC infections because they are intrinsically resistant to most antibiotics, including aminoglycosides, polymyxins B, and colistin [20]. Ceftazidime, meropenem, and co-trimoxazole have been reported as a drug of choice, alone or in combination [21]. Antibiotic resistance may differ according to the center. In a five year-surveillance from Turkey, BCC infections (predominantly pneumonia) reported 50% resistance towards amikacin, carbapenems, cefepime, ciprofloxacin, and co-trimoxazole; 61% resistance was noted towards ceftazidime [22]. In our study, while all agents were sensitive to ceftazidime, meropenem, and levofloxacin, the sensitivity rate of co-trimoxazole was 57.1%. Although there was no statistical difference, the sensitivity to co-trimoxazole varied between 0% and 50% between centers. This may indicate that the agents do not originate from a single center in our study. Since co-trimoxazole and fluoroquinolones are not first-line systemic antibiotics for neonates, ceftazidime and meropenem (in combination or alone) are the most appropriate treatment options for BCC infections in neonates. While treating a case coming from another medical center and for whom sufficient information is unknown, empirical antibiotic therapy should also cover multi-drug-resistant bacteria. Since our center accepted many cases across borders, we had to use meropenem in empirical treatment during this cluster. Switching to a narrower-spectrum antibiotic (i.e., ceftazidime if a BCC is detected) or discontinuing the antibiotic entirely if there is no infection should be considered by the clinician to prevent an increase in antibacterial resistance.

Mortality due to BCC nosocomial infections varies widely between studies. When we look at neonatal studies, it is seen that mortality is reported between 0% and 25% [4,6,7,18,19]. In our study, the mortality rate was 14.3%. The heterogeneity of case groups and lack of further molecular analysis of the strains in most studies may be the main reasons for the considerable variation in mortality rates between studies. Indeed, in our study, the presence of an underlying disease and the need for a mechanical ventilator or inotropic support during culture positivity were associated with increased mortality among infected infants. In an outbreak of *B. cepacia* bacteremia involving 95 adult ICU patients over four years in northern Taiwan, patients with more severe disease at the onset of infection and patients with significant underlying diseases, including malignancy, had higher mortality [23]. Rapidly initiating appropriate antibiotics and complete supportive treatment are critical for improving patient outcomes.

It is known that the epidemiology of infectious diseases changes when access to health services and health resources are limited in war and other conflict environments. Possible increases in the prevalence of vaccine-preventable diseases such as measles and polio in conflict situations such as Syria and recently Ukraine raise concerns all over the world [24,25,26]. Many infectious diseases, such as HIV, antibiotic resistance, multi-resistant tuberculosis, and HCV, are also expected to be affected by mass migration movements that occur under unsuitable conditions [27,28]. Wars always lead to new-onset, conflict-associated, preventable infectious diseases in children and adolescents. However, there are not many studies on this subject related to neonatal infections. *B. cepacia* was identified as the most common cause of early neonatal sepsis in Yemen [7]. A study conducted in a tertiary hospital in Lebanon reported that BCC infections were more common in refugees [8]. These findings raise the concern that countries at war might be at increased risk for such infections.

Relationships between conflicts (in the Middle East [29] and Ukraine [30]) and *Acinetobacter* spp. infections and the possibility of refugees transferring antibiotic resistance away from conflict zones have been the subject of a few different studies [31]. In our study, it is noteworthy that the infection rate due to BCC is high in infants born across the border. Despite the lack of adequate studies on the source of infection in the centers, the high infection rate in neonates due to BCC in conflict zones is very important for both the healthcare institutions in these regions and the centers accepting cases from these regions. More studies should focus on this point when we witness more wars and conflicts in various parts of the world.

Continuous surveillance for BCC and rapid initiation of infection control investigation of any clusters of infection and colonization are crucial for early detection and resolving outbreaks. Strict infection control measures are critical in preventing patients’ acquisition of BCC. If isolating the patients is not feasible, cohorting patients can prevent the spread. Compliance with hand hygiene, proper use of personnel protective equipment, and dedicated medical equipment for the affected patient (e.g., thermometer, stethoscope, and pulse oximeter) are other precautions for infection control [5]. We have successfully prevented cross-contamination within the unit for months with infection control measures.

The most important limitation of our study was that we could not perform formal molecular identification and genotyping. Differentiation of species within the BCC can be particularly problematic as they are phenotypically very similar and most commercial automated bacterial identification systems, such as BD Phoenix^TM^ or VITEK^TM^ 2, cannot reliably distinguish between them [32,33]. Therefore, we could not comment on the clonal relationship between the BCC isolates and determine which subspecies are implicated in this outbreak. Although no further analyses were performed, the isolated bacteria’s antibiotic susceptibilities were similar, which strengthens the possibility that they were from the same subspecies. In the outbreaks reported worldwide, the source has been traced and identified with molecular typing and clonality in some case series. In most of these outbreaks, PCR, ribotyping, and random amplified polymorphic DNA analysis have been used as tools for epidemiological typing [5,34]. Unfortunately, it is not uncommon that identification of BCC species cannot be performed in outbreak analyses. A systematic review of 111 healthcare-associated BCC outbreaks reported that bacteriological typing was not conducted in 4.5% of the outbreaks, and 28.8% of the studies did not mention typing at all [16]. Another limitation was the inability to evaluate maternal and regional risk factors. Data on many variables, such as maternal infections, premature rupture of membranes and chorioamnionitis rates, antibiotic use before and after birth, and the frequency of BCC infections in pediatric and adult cases in the centers, could not be reached. Source research could not be conducted in all but one of the centers. Therefore, we could not make an overall assessment of the origin of clusters and the situation in cross-border centers.

Additionally, detailed information about the medical management of neonates at the birthplace could not be reached, so risk factors related to mortality could be evaluated in a limited framework on a case-by-case basis. On the other hand, this study is one of the largest case series in the literature associated with BCC infection in neonates. Moreover, this is the first study that shows the importance of BCC in neonatal sepsis in areas under war conditions in Syria.

## 5. Conclusions

We showed that BCC is a significant cause of infection and mortality in neonates in conflict zones where access to adequate healthcare is limited. It will be beneficial for centers accepting cases from similar regions to be alert about BCC infections to detect bacteremia early, plan appropriate treatments, and prevent cross-contamination within the unit.

## Figures and Tables

**Figure 1 children-09-01566-f001:**
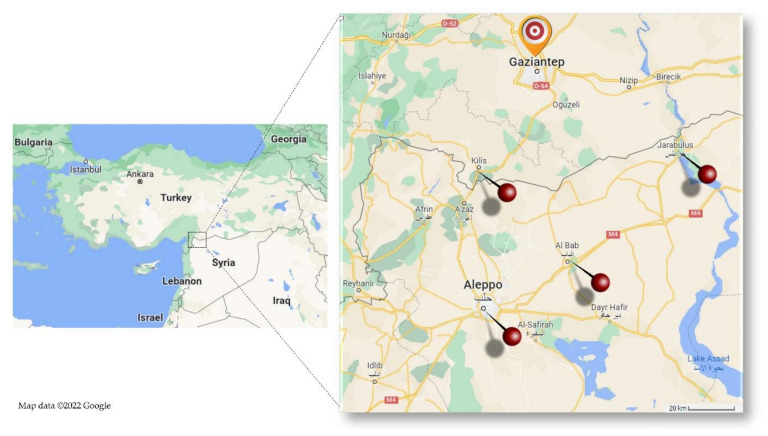
Centers in and outside the country from which the neonates were referred to our hospital in Gaziantep. Google. (n.d.). [Google Maps Gaziantep and Aleppo]. Retrieved from https://goo.gl/maps/m3td1Yo6xU92fRgg7 (accessed on 5 September 2022).

**Figure 2 children-09-01566-f002:**
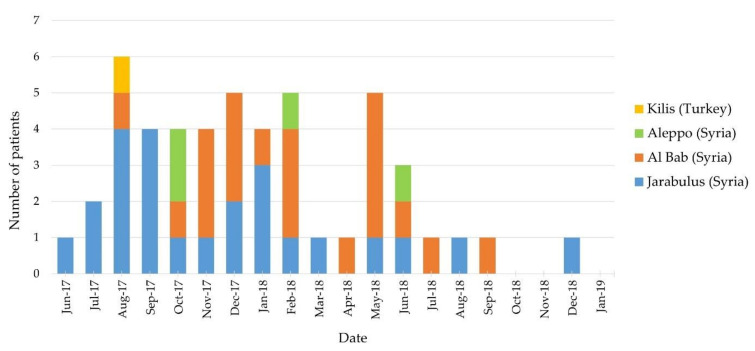
Distribution of the neonates with *Burkholderia cepacia* complex infections regarding birthplace and admission date.

**Table 1 children-09-01566-t001:** Demographic characteristics and initial clinical features of the neonates with *Burkholderia cepacia* complex infections.

Variables	All Cases (*n* = 49)
Age at the admission, day, median (min.–max.)	3 (1–27)
Male, *n* (%)	33 (67.3)
Nationality, *n* (%)	
Turkish	0
Syrian	49 (100)
Birthplace, *n* (%)	
Jarabulus—Syria	24 (49)
Al Bab—Syria	20 (40.8)
Aleppo—Syria	4 (8.2)
Kilis—Turkey	1 (2)
Hospitalization duration at the initial center, hour, median (min.–max.)	11.5 (1–120)
Mode of delivery, *n* (%)	
Cesarean section	20 (40.8)
Vaginal	29 (59.2)
Gestational weeks at delivery, median (min.–max.)	37 (26–39)
Prematurity, *n* (%)	24 (49)
Birth weight, gram, median (min.–max.)	2300 (910–3200)
Birth weight categories, *n* (%)	
NBW	21 (42.9)
LBW	24 (49)
VLBW	3 (6.1)
ELBW	1 (2)
Co-existing problems, *n* (%)	
None	33 (66.7)
Multiple congenital malformations	6 (12.2)
Atresia of the GI tract	4 (8.2)
Congenital heart defects	2 (4.1)
Diaphragmatic hernia	1 (2)
Pneumothorax	1 (2)
ABO incompatibility	1 (2)
Neurological disturbances	1 (2)

Abbreviations: ELBW, extremely low birth weight; GI, gastrointestinal; LBW, low birth weight; NBW, normal birth weight; VLBW, very low birth weight.

**Table 2 children-09-01566-t002:** Microbiological characteristics, laboratory values, clinical profile during the infection, and outcomes of the neonates with *Burkholderia cepacia* complex infections.

Variables	All Cases (*n* = 49)
Sample from which organism isolated, *n* (%)	
Blood	48 (98)
Blood + CSF	1 (2)
Antibiotic susceptibility, *n* (%)	
Ceftazidime	49 (100)
Co-trimoxazole	28 (57.1)
Levofloxacin	49 (100)
Meropenem	49 (100)
Time from admission to our center to the collection of the positive sample, *n* (%)	
<24 h	44 (89.8)
24–<48 h	2 (4.1)
48–<72 h	3 (6.1)
Central catheter use, *n* (%)	23 (46.9)
TPN support, *n* (%)	16 (32.7)
Respiratory support, *n* (%)	
None	11 (22.4)
Noninvasive (CPAP/NIMV)	14 (28.6)
Invasive MV (SIMV)	24 (49)
Need for inotrope infusion, *n* (%)	17 (34.7)
Laboratory values, median (min.–max.)	
WBC, cells/mm^3^	12,190 (2710–46,870)
ANC, cells/mm^3^	6890 (200–39,510)
ALC, cells/mm^3^	2640 (180–9870)
Platelets, cells/mm^3^	86,000 (2000–318,000)
CRP, mg/L	50 (1–248)
Procalcitonin, ng/mL	1.8 (0.2–100)
Elevated CRP (>5 mg/L), *n* (%)	42 (85.7)
Elevated procalcitonin (>0.5 ng/mL), *n* (%)	36 (73.5)
Outcome	
Lenght of stay, day, median (min.–max.)	25 (1–180)
Mortality, *n* (%)	7 (14.3)

Abbreviations: ALC, absolute lymphocyte count; ANC, absolute neutrophil count; CPAP, continuous positive airway pressure; CRP, C-reactive protein; CSF, cerebrospinal fluid; MV, mechanical ventilation; NIMV, nasal intermittent mechanic ventilation; SIMV, synchronized intermittent mechanical ventilation; TPN, total parenteral nutrition; WBC, white blood cell.

**Table 3 children-09-01566-t003:** Characteristics of the non-survivor neonates with *Burkholderia cepacia* complex infections.

Case No.	Birthplace	MOD	GA (Week)	BW	Gender	Age at Admis. (Day)	Underlying Diseases	Respir. Support	Need of Inotrope Infusion	ANC (cells/mm^3^)	ALC (cells/mm^3^)	Plt (cells/mm^3^)	CRP (mg/L)	PCT (ng/mL)	Clinical Course	LOS (day)
1	Jarabulus	C/S	34	LBW	Female	15	None	SIMV	Yes	6890	4960	2000	248.0	12.0	Septic shock, DIC, MODS	13
2	Al Bab	C/S	38	NBW	Male	2	None	SIMV	Yes	2350	2340	14,000	97.0	100.0	Septic shock, DIC, MODS	3
3	Al Bab	VD	39	NBW	Male	1	Diaphragmatic hernia	SIMV	Yes	16,400	4710	240,000	6.0	0.5	Asphyxia, MODS	7
4	Jarabulus	C/S	39	LBW	Male	1	Ventricular septal defect, pulmoner atresia	SIMV	No	11,320	8740	77,000	4.5	3.0	Cardiac insufficiency, MODS	3
5	Al Bab	VD	37	LBW	Female	3	Anal atresia	SIMV	Yes	1560	780	318,000	10.9	2.4	Post-operation sepsis	3
6	Jarabulus	C/S	38	NBW	Female	7	Multiple congenital malformations	SIMV	Yes	3240	2470	8000	81.0	2.4	*Klebsiella spp.* co-infection, septic shock, DIC	15
7	Al Bab	VD	38	Normal	Female	11	Double outlet right ventricle, arcus aorta hypoplasia, Trisomy 18	SIMV	Yes	4280	5040	5000	75.6	3.2	Cardiac insufficiency, sepsis	5

Abbreviations: ALC, absolute lymphocyte count; ANC, absolute neutrophil count; BW, birth weight; C/S, caesarean section; CRP, C-reactive protein; DIC, disseminated intravascular coagulation; GA, gestational age; LBW, low birth weight; LOS, length of stay; MOD, mode of delivery; NBW, normal birth weight; SIMV, synchronized intermittent mechanical ventilation; Plt, platelet; VD, vaginal delivery.

**Table 4 children-09-01566-t004:** Comparison of the variables between survivor and non-survivor neonates with *Burkholderia cepacia* complex infections.

Variables	Survivors (*n* = 42)	Non-Survivors (*n* = 7)	*p*
Age at the admission, day, median (min.–max.)	3 (1–27)	3 (1–15)	0.75
Male, *n* (%)	29 (69)	4 (57.1)	0.53
Birthplace, *n* (%)			0.69
Jarabulus—Syria	21 (50)	3 (42.9)	
Al Bab—Syria	16 (38.1)	4 (57.1)	
Aleppo—Syria	4 (9.5)	0	
Kilis—Turkey	1 (2.4)	0	
Mode of delivery, *n* (%)			0.34
Cesarean section	16 (38.1)	4 (57.1)	
Vaginal	26 (61.9)	3 (42.9)	
Gestational weeks, median (min.–max.)	36 (26–39)	38 (34–39)	0.05
Prematurity, *n* (%)	23 (54.8)	1 (14.3)	0.05
Birth weight, gram, median (min.–max.)	2200 (910–3200)	2630 (1500–3200)	0.16
Birth weight categories, *n* (%)			0.77
NBW	17 (40.5)	4 (57.1)	
LBW	21 (50)	3 (42.9)	
VLBW	3 (7.1)	0	
ELBW	1 (2.4)	0	
Co-existing problems, *n* (%)			0.02
No	31 (73.8)	2 (28.6)	
Yes	11 (26.2)	5 (71.4)	
Central catheter use, *n* (%)	19 (45.2)	4 (57.1)	0.56
TPN support, *n* (%)	12 (28.6)	4 (57.1)	0.14
Respiratory support, *n* (%)			0.01
None	11 (26.2)	0	
Noninvasive (CPAP/NIMV)	14 (33.3)	0	
Invasive MV (SIMV)	17 (40.5)	7 (100)	
Need for inotrope infusion, *n* (%)	11 (26.2)	6 (85.7)	0.002
Elevated CRP (>5 mg/L), *n* (%)	36 (85.7)	6 (85.7)	1.0
Elevated procalcitonin (>0.5 ng/mL), *n* (%)	30 (71.4)	6 (85.7)	0.42
Co-trimoxazole resistance, *n* (%)	19 (45.2)	5 (28.6)	0.41
Time from admission to our center to the collection of the positive sample, *n* (%)			0.62
<24 h	37 (88.1)	7 (100)	
<4–<48 h	2 (4.8)	0	
48–<72 h	3 (7.1)	0	
Laboratory values, median (min.–max.)			
WBC, cells/mm^3^	11,840 (2710–46,870)	12,430 (3520–23,680)	0.88
ANC, cells/mm^3^	7050 (200–39,510)	4280 (1560–16,400)	0.38
ALC, cells/mm^3^	2600 (180–9870)	4710 (780–8740)	0.23
Platelets, cells/mm^3^	88,000 (7000–314,000)	14,000 (2000–318,000)	0.20
CRP, mg/L	48 (1–240)	75.6 (4.5–248)	0.73
Procalcitonin, ng/mL	1.1 (0.2–100)	3 (0.5- 100)	0.14
Neutrophil to lymphocyte ratio	2.3 (0.1–136.7)	1.3 (0.8–3.5)	0.06

Abbreviations: ALC, absolute lymphocyte count; ANC, absolute neutrophil count; CPAP, continuous positive airway pressure; CRP, C-reactive protein; ELBW, extremely low birth weight; LBW, low birth weight; MV, mechanical ventilation; NBW, normal birth weight; NIMV, nasal intermittent mechanic ventilation; SIMV, synchronized intermittent mechanical ventilation; TPN, total parenteral nutrition; VLBW, very low birth weight; WBC, white blood cell.

## Data Availability

The data presented in this study are available on request from the corresponding author.

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
