# Peer review of "Burkholderia cepacia Complex Infections in Urgently Referred Neonates from Syrian Border Regions to a Hospital in Turkey: A Cross-Border Cluster"

_children, 2022, doi:10.3390/children9101566_

Round 1

Reviewer 1 Report

General comments.

The findings reported here are of potential interest to the medical community. The main weakness of the study is the absence of molecular typing (such as MLST or WGS) that would allow for a determination of the clonal relationship between the BCC isolates and also determine which sub-species are implicated in this outbreak. If the isolates remain available in storage I would strongly encourage collaboration with a specialist laboratory to type the available isolates.

Without such data, the paper is still of interest. However, I feel that more information could be provided on the antibiotics that were used for successful treatment of neonates with BCC bacteraemia, once the blood cultures results were determined. If it is possible to distinguish successful treatments from unsuccessful treatments using statistical methods that would also be very valuable to readers. A recommendation of what the authors would recommend for successful treatment (even if not statistically proven) is valuable for other doctors who may be exposed to similar outbreaks in the future. Too little is said about antibiotic treatment.

Introduction

The Burkholderia cepacia complex (BCC) is widely recognised as a group of subspecies or genomovars, of which genomovar 1 is named Burkholderia cepacia.  There are thought to be at least 22 closely related species within the complex (see Jin et al. Biol. Direct 2020, 15, 6.). When discussing general characteristics in the introduction, the authors should refer to the Burkholderia cepacia complex. The authors should not refer to “B. cepacia” anywhere in the paper unless they are specifically referring to genomovar 1 of the BCC. In the absence of genotyping, the authors should refer to BCC throughout their paper.

Line 63: Please provide a reference for the CLSI documents used.

Line 73: Mac Conkeys should be MacConkey agar.

Line 121: Figure 2, title. If these are neonates with BCC positive cultures, please say so in the title.

Line 147: I cannot see any CLSI interpretive criteria for ciprofloxacin in the latest relevant CLSI document (Table 2B-3, CLSI M100, 32ND Edition). How was ciprofloxacin susceptibility interpreted?

Line 161: use italics for bacterial names. Check throughout.

Line 207: What are swap cultures? Do you mean swab cultures?

Line 212: Correct the spelling of Klebsiella pneumoniae. Contaminated ultrasound gel is known to be a source of transmission for BCC. Was any cultured?

Reviewer 2 Report

I would like to kindly thank the authors for submitting this very important data for consideration of publication. However, without formal molecular identification of your strains of bacteria it is not possible to confirm that all these cases of sepsis were due to B. cepacia. Conventional biochemical methods are not 100% reliable for B. cepacia identification.

For your consideration:

1.       Were all these strains resistant to colistin by broth microdilution or grew well on selective media for B. cepacia isolation?

2.       Members of the Burkholderia cepacia complex can be confirmed by simple recA PCR, though subspeciation requires sequencing of this and other genes. Confirmation by recA PCR could be sufficient for this study.

3.       Were any other antimicrobials tested and how did you interpret the results?

4.       I’m interested to know which antibiotics these patients received, and did they combat the infections?

Round 2

Reviewer 1 Report

None.

Reviewer 2 Report

Thank you very much for your revised manuscript.  Your study provides valuable information that could save the lives of sick babies by being aware of the risk of BCC infection, nevertheless molecular confirmation should be sought as non-molecular methods can be inaccurate for BCC.